



# On the Uncertainty of Digital Twin Models for Load Monitoring and Fatigue Assessment in Wind Turbine Drivetrains

Felix C. Mehlan[1] and Amir R. Nejad[1]

[1]Norwegian University of Science and Technology, Jonsvannsveien 82, 7050 Trondheim, Norway

**Correspondence:** Felix C. Mehlan (felix.c.mehlan@ntnu.no)

**Abstract.** This article presents a systematic assessment of the uncertainty in digital twins for load and fatigue monitoring in wind turbine drivetrains. The uncertainty in the measurement input, the reduced order drivetrain models and the model updating methods are investigated. A statistical analysis is conducted on gear and bearing load measurements from numerical studies with 5 and 10 MW drivetrain models and from field measurements of a 1.5 MW research turbine. The uncertainty is quantified using log-normal distributions and limitations of digital twin are discussed such as the measurement uncertainty in 10 min averaged SCADA data, the uncertainty in estimating the unknown rotor torque, and the modelling errors in torsional reduced order drivetrain models. This study contributes to a deeper understanding of the origin and the effects of uncertainty in digital twins and delivers a foundation for further reliability and risk assessment studies.

## 1 Introduction

Offshore wind turbine installations are projected to accelerate rapidly in the near future driven by better wind resources and higher social acceptance rates compared to onshore sites (Wind Europe, 2020). However, a major economic limitation of offshore wind turbines are high operational and maintenance expenditures (OPEX), which amount to about 34 % of the levelized cost of energy (LCOE) (Stehly and Beiter, 2020). These are caused by lower reliability due to harsher environmental conditions and time-consuming replacement or repair due to difficulties accessing the site and dependency on good weather conditions. A major contributor to the OPEX is the geared drivetrain with frequent failures and long downtimes and is thus the subject of current research (Wilkinson et al.).

Digital twin (DT) is an emerging technology with prospects of decreasing the OPEX and improving the market competitiveness of offshore wind farms. The wind turbine drivetrain DT proposed by the authors in (Mehlan et al., 2022a) would enable monitoring drivetrain loads and fatigue damage at otherwise inaccessible locations such as bearing and gear contacts using "virtual sensors". A DT framework with the three components DT Data, DT Model and DT Decision support is envisioned for this objective (Fig. 1). The *DT Data* comprise continuous data streams provided by the supervisory control and data acquisition system (SCADA) and the condition monitoring system (CMS), the data history including the load history and the accumulated fatigue damage, asset information such as the drivetrain topology, and general domain knowledge on drivetrain physics. The *DT Model* refers to physics-based models to simulate internal drivetrain dynamics. Reduced order models (ROMs) are derived from high-fidelity multibody simulation (MBS) models that are considered full-order models (FOMs) for the purpose of





real-time simulation. The virtual model and its physical counterpart are synchronized with real-time field measurements using model updating techniques. State estimators such as Kalman filters are applied to infer the dynamic states of the drivetrain at small time intervals, given by the sensor sample frequency of 200 Hz. System identification methods are used to estimate system parameters such as inertia, stiffness and damping parameters, as a means to validate values provided by gearbox man-
ufacturers or to track long-term parameter variations due to faults, material degradation or other mechanisms. Therefore it is sufficient to update the model parameters at longer time intervals, here set to 10 min. The model updating, also referred to as data fusion or digital twinning, is essential as it facilitates the use of virtual sensors in the synchronized model. The virtual sensor measurements are converted to value-adding information for the turbine operator in the component called *DT Decision support*. The focus lies on long-term fatigue damage and remaining useful life (RUL) assessment of drivetrain components,
which is necessary to advance from corrective to predictive maintenance strategies.

In previous numerical and field studies the proof of concept of the DT framework could be demonstrated (Mehlan et al., 2022a)(Mehlan et al., 2023), however, there remain research questions on the sources and the magnitude of the the virtual measurements' uncertainty. Uncertainty is present in the DT's data input due to the stochastic nature of wind and wave loads, as well as in the load and fatigue calculations due to the limitations of the DT model.

The uncertainty in long-term fatigue damage calculation of wind turbine drivetrains is addressed in several studies on reliability-based design (Nejad et al., 2014)(Li et al., 2017)(Dong et al., 2020). Nejad et al. presents a method for fatigue analysis for gear tooth root bending and differentiate between the uncertainty in the aeroelastic model, the drivetrain model and the fatigue damage model (Nejad et al., 2014). The uncertainty is characterized by log-normal distributions with standard deviation values ranging from 0.01 for the drivetrain model to 0.1 for the aeroelastic model. Li et al. present a study on reliability-based design
optimization of gear profiles and consider the uncertainty of the wind conditions with a joint probability density function of the wind speed and turbulence intensity (Li et al., 2017). Dong et al. further consider model uncertainties in a wide range of drivetrain and fatigue model parameters (Dong et al., 2020).

The aforementioned studies are focused on the design of wind turbine drivetrains, where the aleatory uncertainty in the unknown environmental conditions is most influential. For DTs of operating wind turbines the challenge shifts from aleatory
uncertainty towards epistemic uncertainty, since the environmental conditions and the dynamic system response are continuously estimated using real-time measurements and state estimation methods. The epistemic uncertainty of such methods has not yet been investigated systematically, as this approach is relatively novel in the field of wind energy. The presented study bridges this gap and contributes to a better knowledge of the origin, the magnitude and the distribution shape of the uncertainty in DTs for fatigue damage monitoring.

The remainder of this article is structured as follows: Sec. 2 presents the methodology of the DT framework for fatigue damage monitoring and defines the numerical and experimental case studies for uncertainty assessment. Sec. 3 discusses the uncertainty in different DT components and their impact on long-term fatigue damage. Concluding remarks are provided in Sec. 4.





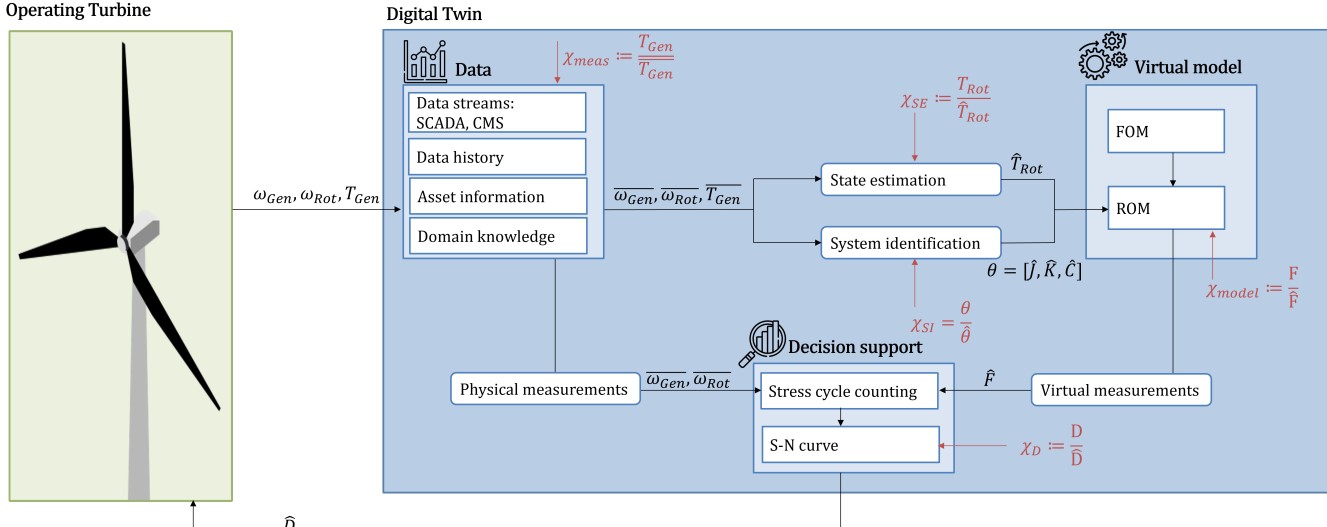

**Figure 1.** Digital twin framework for continuous remaining useful life estimation in wind turbine drivetrain components and sources of uncertainty (Mehlan et al., 2022a).

## 2 Methodology

### 2.1 Definition of uncertainty

The proposed DT framework comprises several interacting models and data processing algorithms, each of which introduce characteristic uncertainties. These uncertainties are grouped into the categories of measurement, state estimation, system identification, model and fatigue damage uncertainty. The measurement uncertainty $\chi_{meas}$ originates from poor sensor data quality due to measurement noise, sensor failure or the low sampling frequency, which imposes a frequency limit on the observable drivetrain load spectrum. Only the latter is investigated in this study, since simulation measurements are used. The measure-

ment uncertainty is defined here as the ratio of the true generator torque $T_{Gen}$ to the measured generator torque $\bar{T}_{Gen}$. The true generator torque is sampled from simulation measurements at 200 Hz, which is sufficient to reflect all relevant drivetrain dynamics, while the measured generator torque is obtained by averaging the simulation measurements in 1 s or 10 min intervals, which is the typical resolution of SCADA data.

$$\chi_{meas} := \frac{T_{Gen}}{\bar{T}_{Gen}} \tag{1}$$

The state estimation uncertainty $\chi_{SE}$ refers to errors caused by the Kalman filter algorithm. The Kalman filter fuses uncertain information from measurements and model predictions and is the optimal state estimator in case of white Gaussian measurement and process noise. However, this assumption is not valid here since the unknown rotor torque modelled as process noise exhibits non-uniformity such as peaks at characteristic excitation frequencies (1P, 3P, ...). It is therefore expected that use of Kalman filter introduces an additional uncertainty in the drivetrain case. This uncertainty is defined as the ratio of the true





dynamic states $x$ to the states estimated by Kalman filtering $\hat{x}$.

$$\chi_{SE} := \frac{x}{\hat{x}} \tag{2}$$

The system identification uncertainty $\chi_{SI}$ reflects the error that is introduced by the inverse methods to estimate the system's inertia, stiffness and damping matrices $\hat{J}, \hat{K}, \hat{C}$ and is defined as the ratio of the true system parameters $\theta$ to the estimated parameter set $\hat{\theta}$.

$$\chi_{SI} := \frac{\theta}{\hat{\theta}} \tag{3}$$

The model uncertainty $\chi_{model}$ refers to to the limitations of the DT model to simulate all relevant drivetrain dynamics. ROMs with a limited number of torsional DOFs are considered, which are unable to capture non-torsional drivetrain dynamics such as shaft bending modes or complex torsional dynamics such as gear meshing. The error caused by the model complexity reduction is described with the model uncertainty and defined as the ratio of the drivetrain loads $F, \hat{F}$ calculated with the FOM and the

ROM, respectively

$$\chi_{model} := \frac{F}{\hat{F}} \tag{4}$$

The uncertainty of the fatigue damage model, $\chi_D$ as shown in Fig. 1, including the stress cycle counting method and the S-N curves is related to the material and fatigue testing which is out the scope of this numerical case study.

## 2.2    Numerical case studies

Two numerical case studies with the National Renewable Energy Laboratory (NREL) 5 MW baseline turbine (Jonkman et al., 2009) and the DTU 10 MW reference turbine are conducted (Bak et al., 2013). The best practice for dynamic drivetrain simulation is the decoupled analysis approach, where the "global", structural blade and tower dynamics and the "internal" drivetrain dynamics are simulated separately (Nejad et al., 2014). The global system response is simulated first with an aeroelastic model

and the resultant main shaft loads and nacelle motions are then imposed as boundary conditions on the drivetrain model. This procedure is motivated by the fact that the global dynamics are governed by aerodynamic excitations and occur at low frequencies ($< 10Hz$), while much higher frequencies such as gear meshing frequencies at $> 100Hz$ need to be considered for the drivetrain dynamics. The simulation cases are designed according to the IEC 61400-1 requirements for long-term fatigue analysis. Twelve wind speed cases ranging from cut-in wind speed of 3 m/s to cut-out wind speed of 25 m/s are considered.

One case of turbulence intensity is considered and modelled with the IEC turbulence classes A. Only one case of wave height and wave period are considered, since the drivetrain bearing and gear loads are reportedly insensitive to the sea state. The primary effect of harsher sea states can be observed in increased axial loads induced by pitch motions, which are compensated by the main bearings in a four-point suspension and do not propagate further into the drivetrain (Nejad et al., 2015). Each environmental condition (EC) is simulated for one hour with six different random realizations (seeds) of turbulent wind fields.





**Table 1.** Environmental conditions for simulation with global and drivetrain models.

| Wind speed [m/s] | 3...25 |
|---|---|
| Turbulence intensity [-] | IEC class A |
| Wave height [m] | 5 |
| Wave period [s] | 12 |
| Simulation length [s] | $6 \times 3600$ |

## 2.3 Global models

The global wind turbine dynamics are simulated with open source aeroelastic models of the NREL 5 MW and DTU 10 MW reference turbines mounted on the OC4 and Nautilus semisubmersible platforms, respectively (Robertson et al., 2012)(Arias and Galvan, 2018). The models are implemented in the aeroelastic code OpenFAST that comprises of computational modules for calculation of the aerodynamics, hydrodynamics, structural dynamics and wind turbine control (OpenFAST, 2022). The aerodynamics are calculated with blade element momentum (BEM) theory, where the turbulent wind field is generated with the Kaimal turbulence model. The structural dynamics of the blades and the tower are based on Timoshenko elastic beam theory. The incident wave loads on the floater are modelled with a Jonswap spectrum. A variable-speed controller is implemented for the 5 MW and the 10 MW model.

## 2.4 Full order drivetrain models

Multibody simulation (MBS) models of the NREL 5 MW and DTU 10 MW reference turbine serve as benchmark in this study (Nejad et al., 2016)(Wang et al., 2020). The MBS models are developed according to best practices and current model fidelity guidelines (Guo et al., 2015) and are thus considered as FOMs. Both FOMs have similar topology and comprise a four-point suspension for the main shaft and a gearbox with two planetary gear stages and one parallel gear stage (Fig. 2). However, the 5 MW model represents a high-speed gearbox with a gear ratio of 1:96.354, while the 10 MW model represents a medium-speed gearbox with a gear ratio of 1:50.039. The FOMs allow shaft motion in all six degrees of freedom (DOF) and consider the flexibility in the main shaft and the planet carriers. The bearings and the torque arm bushings are modelled as linear spring-damper connections in six DOF with diagonal stiffness and damping matrices. The gear compliance is modelled with a time-invariant mesh stiffness function capable of emulating gear meshing excitations. The input loads simulated with aeroelastic models are imposed on the main shaft, while the generator shaft speed is controlled with a PI-controller.

## 2.5 Reduced order drivetrain models

Reduced order models (ROMs) are preferable as DT models due to the high computational costs in real-time monitoring applications (Mehlan et al., 2022b). The complexity of DT models is also limited by the observability requirement of the state estimator. The state estimator that is used to match the dynamics of the DT model with the physical turbine requires that all dynamic states are observable with the available measurement input. The SCADA measurements of the main and generator



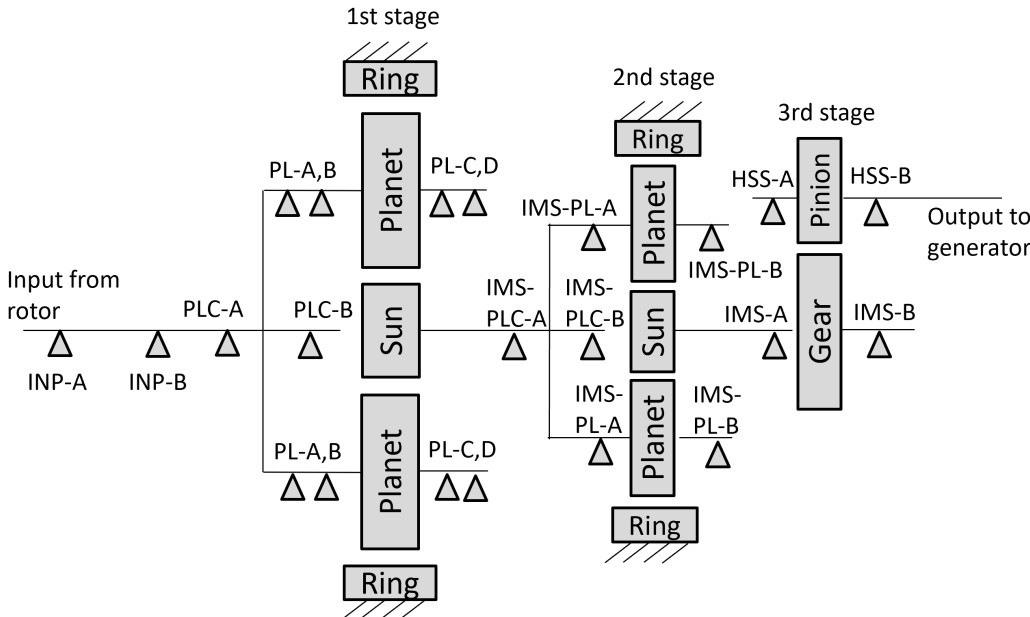

**Figure 2.** Topology and component nomenclature of the NREL 5MW and DTU 10MW drivetrain models.

shaft speeds allow the observation of torsional drivetrain modes. Bending and lateral drivetrain modes are observable with CMS accelerometers mounted on the gearbox housing, however the sensitivity is relatively low due to measurement noise and the observation function is complex due to the transfer path of the vibration through the housing (Mehlan et al., 2022b). For this reason, the ROMs are limited to torsional degrees of freedom (DOF) only. Lumped parameter models with one and two torsional DOFs are considered. The input torques at each gear stage $T_{in,k}$ are calculated with the torsional ROMs and then further used to determine local gear and bearing forces (Sec. 2.5.3).

### 2.5.1 Rigid one degree of freedom ROM

The first ROM represents a rigid, torsional model with one degree of freedom (DOF). The flexibility of shafts and gear contacts are neglected, which yields direct coupling of the angular shaft velocities $\omega_k$ and input torques at each gear stage $T_{in,k}$ via the gear ratios $i_k$

$$\omega_{Rot} = \omega_{in,2}/i_1 = \omega_{in,3}/i_1/i_2 = \omega_{Gen}/i_1/i_2/i_3$$

$$T_{in,1} = i_1 T_{in,2} = i_1 i_2 T_{in,3} = i_1 i_2 i_3 T_{Gen} \tag{5}$$

The rigid ROM is advantageous, in that it does not require inertia, stiffness or damping parameters for model construction and validation, which minimizes the uncertainty associated with system identification techniques for parameter estimation ($\chi_{SI}$). In addition, it is not necessary to apply state estimation methods, since the gear stage torques and thus all drivetrain loads are



directly observable with the measured generator torque, which reduces state estimation uncertainties ($\chi_{SE}$).


### 2.5.2 Flexible two degree of freedom ROM

The second ROM introduces one additional torsional DOF and is able to represent the first torsional mode. However, this model assumes knowledge of inertia, stiffness and damping parameters, which may be estimated via system identification techniques. The flexibility of all drivetrain components are lumped into a scalar drivetrain stiffness $k_{DT}$, while the torsional inertias are

lumped into either the rotor inertia $J_{Rot}$ or the generator inertia $J_{Gen}$. The equations of motion are then given by

$$\mathbf{J}\ddot{\phi} + \mathbf{C}\dot{\phi} + \mathbf{K}\phi + \mathbf{f} = \mathbf{0} \tag{6}$$

where $\mathbf{J}$ denotes the inertia matrix, $\mathbf{C}$ is the damping matrix, $\mathbf{K}$ is the stiffness matrix, $\mathbf{f}$ is the external force vector and $\phi$ are the independent dynamic states

$$\mathbf{J} = \begin{bmatrix} J_{Rot} & 0 \\ 0 & J_{Gen} \end{bmatrix}, \mathbf{C} = \begin{bmatrix} c_{DT} & -c_{DT}/i_{DT} \\ -c_{DT}/i_{DT} & c_{DT}/i_{DT}^2 \end{bmatrix}, \mathbf{K} = \begin{bmatrix} k_{DT} & -k_{DT}/i_{DT} \\ -k_{DT}/i_{DT} & k_{DT}/i_{DT}^2 \end{bmatrix}$$

$$\phi = \begin{bmatrix} \phi_{Rot} \\ \phi_{Gen} \end{bmatrix}, \mathbf{f} = \begin{bmatrix} -T_{Rot} \\ T_{Gen} \end{bmatrix}, \tag{7}$$

The gear stage input torques are still coupled and only a function of the rotor and generator shaft angular positions $\phi$

$$T_{in,1} = i_1 T_{in,2} = i_1 i_2 T_{in,3} = [c_{DT}, \ -c_{DT}/i_{DT}]\dot{\phi} + [k_{DT}, \ -k_{DT}/i_{DT}]\phi \tag{8}$$

### 2.5.3 Bearing and gear forces

The gear forces are determined with free body diagrams and moment balances as a function of the gear stage input torques. Dynamic effects of planet load sharing are not considered at the planetary gear stages, hence the gear stage torque is distributed

equally among the number of planets $N_{PL}$. Furthermore, the gear forces at the ring-planet and the sun-planet contacts are assumed to be equal. The circumferential (z-direction) gear forces $F_t$ are then obtained as follows

$$F_{t,1} = T_{in,1} \cdot i_1 / r_{b,S,1} / N_{PL,1}$$

$$F_{t,2} = T_{in,2} \cdot i_2 / r_{b,S,2} / N_{PL,2}$$

$$F_{t,3} = T_{in,3} / r_{b,G,3} \tag{9}$$

where $r_b$ are the base radii of the first and second stage sun and of the third stage gear wheel. The remaining gear force components in x- and y-direction, the axial and radial gear force components $F_a$, $F_r$, are determined with the tangential

pressure angle $\alpha_t$ and helix angle $\beta$. The planetary gear stage is modelled with spur gears ($\beta = 0$), while the parallel gear stage is modelled with a helix angle of $\beta = 10°$

$$F_r = F_t \tan(\alpha_t)/\cos(\beta)$$

$$F_a = F_t \tan(\beta) \tag{10}$$





At the planetary gear stages the radial bearing forces $F_{rad}$ are directly proportional to the circumferential gear forces with the assumption of negligible gravity forces.

$$F_{rad,PL-A} = 2 \cdot F_{t,1}$$

$$F_{rad,IMS-PL-A} = 2 \cdot F_{t,2} \tag{11}$$

At the helical gear stage the radial bearing forces are derived with moment balances

$$F_{rad} = \sqrt{F_y^2 + F_z^2} \tag{12}$$

where

$$F_{y,IMS-A} = -F_r \frac{d_{IMS-B} - d_W}{d_{IMS-B} - d_{IMS-A}} + F_a \frac{r_{p,W}}{d_{IMS-B} - d_{IMS-A}}$$
$$F_{y,IMS-B} = -F_r \frac{d_W - d_{IMS-A}}{d_{IMS-B} - d_{IMS-A}} - F_a \frac{r_{p,W}}{d_{IMS-B} - d_{IMS-A}}$$
$$F_{y,HSS-A} = F_r \frac{d_{HSS-B} - d_P}{d_{HSS-B} - d_{HSS-A}} + F_a \frac{r_{p,P}}{d_{HSS-B} - d_{HSS-A}}$$
$$F_{y,HSS-B} = F_r \frac{d_P - d_{HSS-A}}{d_{HSS-B} - d_{HSS-A}} - F_a \frac{r_{p,P}}{d_{HSS-B} - d_{HSS-A}} \tag{13}$$

175

$$F_{z,IMS-A} = -F_{t,3} \frac{d_{IMS-B} - d_W}{d_{IMS-B} - d_{IMS-A}}$$
$$F_{z,IMS-B} = -F_{t,3} \frac{d_W - d_{IMS-A}}{d_{IMS-B} - d_{IMS-A}}$$
$$F_{z,HSS-A} = F_{t,3} \frac{d_{HSS-B} - d_P}{d_{HSS-B} - d_{HSS-A}}$$
$$F_{z,HSS-B} = F_{t,3} \frac{d_P - d_{HSS-A}}{d_{HSS-B} - d_{HSS-A}} \tag{14}$$

The axial gear force component of the helical high-speed gear stage is supported by the HSS-B and IMS-B bearings.

$$F_{ax,IMS-B} = F_a$$
$$F_{ax,HSS-B} = -F_a \tag{15}$$

## 2.6 Experimental case study

The simulation measurements are partially validated with field measurements of the department of energy (DOE) 1.5 MW research turbine located at the National Renewable Energy Laboratory (NREL) (Santos and van Dam, 2015). The DOE 1.5 MW turbine is equipped with a commercial Winergy PEAB 4410.4 high-speed gearbox with similar three stage topology as the above simulation models. The dataset, originally collected for the analysis of cage and roller slip in the HSS-A bearing (Guo



WIND
ENERGY
SCIENCE
DISCUSSIONS

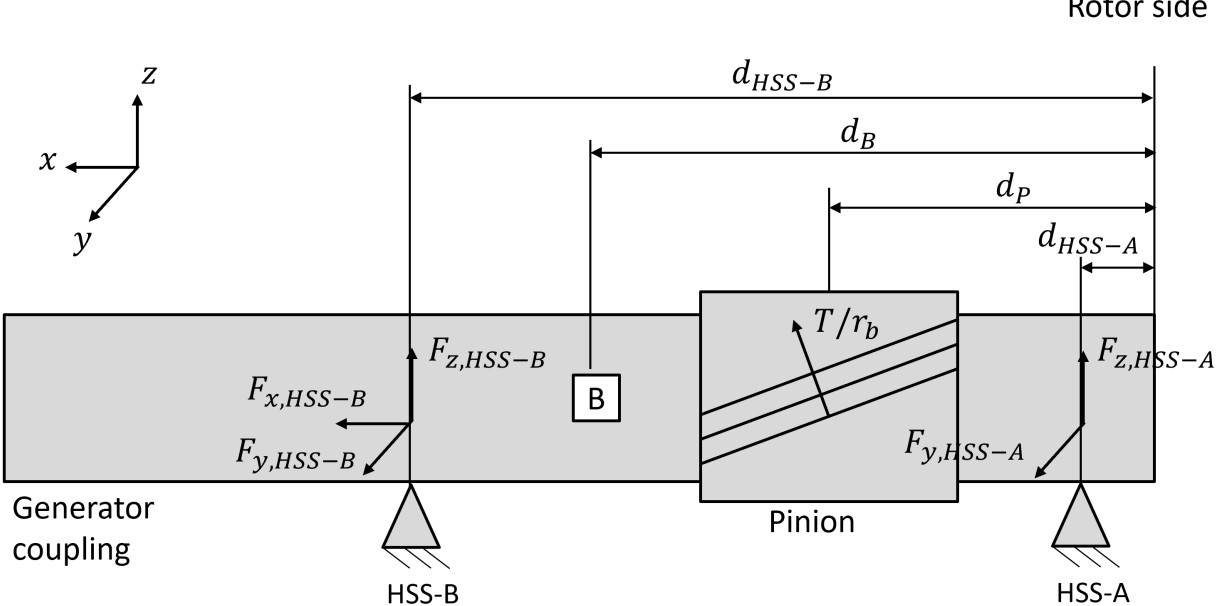

**Figure 3.** Forces at the HSS.

and Keller, 2020), is repurposed for this study. The original sample frequency of 5 kHz necessary to observe slip dynamics
restricted the measurement duration and as a result the total recorded data amounts to only about 30 min. Nonetheless, the full
wind spectrum is covered, which allows for comparison with simulated data.

The loading of the HSS is fully determined with three shaft mounted strain gauge bridges, one for measuring torque and two
90-degree offset bridges for measuring bending moments. The forces at the HSS-A bearing are calculated with the torque $T$
and bending moment measurements $M_y, M_z$ (Guo and Keller, 2020)

$$F_{y,HSS-A} = \frac{1}{d_B - d_{HSS-A}} [-M_z - T/r_b(d_B - d_P)\sin\beta]$$

$$F_{z,HSS-A} = \frac{1}{d_B - d_{HSS-A}} [-M_y - T/r_b(d_B - d_P)\cos\beta] \tag{16}$$

These measurements are considered FOM bearing load measurements, since all relevant torsional and shaft bending dynamics
are captured. The FOM loads are set in relation to the ROM loads calculated solely with torque measurements and the rigid
ROM (eq. 14) to assess the model uncertainty.

## 2.7 State and input estimation

The DT model is synchronized with the operating wind turbine at regular time intervals $\Delta t$ such that gear and bearing loads can
be measured with "virtual sensors" in the synchronized model. The challenge lies in the incomplete and noisy measurements
of both the dynamic states and the input forces, which poses a joint state and input estimation problem. The measurements of





the dynamic states, the shaft angular velocities and positions, are corrupted with measurement noise, while the input forces at the main shaft are unknown; only the generator side torque is measured. The augmented Kalman filter is applied here as

an joint state and input estimator, as it is the optimal estimator for dynamic systems governed by linear, stochastic equations subjected to white Gaussian process and measurement noise. For this purpose the equations of motion of the flexible ROMs are first brought into discrete state-space representation

$$\mathbf{x_{n+1}} = \mathbf{F^d} \mathbf{x_n} + \mathbf{G_k^d} \mathbf{u_{k,n}} + \mathbf{G_u^d} \mathbf{u_{u,n}} + \mathbf{w_n}, \tag{17}$$

$$\mathbf{y_n} = \mathbf{H^d} \mathbf{x_n} + \mathbf{v_n}, \tag{18}$$

where the state vector $\mathbf{x}$ is obtained by stacking the shaft angular positions and velocities, the input forces $\mathbf{u}$ are split into the known generator torque $\mathbf{u_k}$ and the unknown rotor torque $\mathbf{u_u}$, the measurement vector $\mathbf{y}$ contains the rotor and generator shaft speeds, the unknown dynamic component of the rotor torque is considered white Gaussian process noise $\mathbf{w}$ with covariance $\mathbf{Q}$, and $\mathbf{v}$ is white Gaussian measurement noise with covariance $\mathbf{R}$

$$
\begin{aligned}
\mathbf{x} &:= [\phi \ \dot{\phi}]^{\mathrm{T}} \\
\mathbf{u_k} &:= T_{Gen} \\
\mathbf{u_u} &:= T_{Rot} \\
\mathbf{y} &:= [\dot{\phi}_{Rot}, \dot{\phi}_{Gen}]^{\mathrm{T}} \\
\mathbf{w} &\sim \mathcal{N}(\mathbf{0}, \mathbf{Q}) \\
\mathbf{v} &\sim \mathcal{N}(\mathbf{0}, \mathbf{R})
\end{aligned} \tag{19}
$$

The system matrix $\mathbf{F^d}$, the input matrix $\mathbf{G^d}$ and the observation matrix $\mathbf{H^d}$ of the discrete state-space model are calculated as follows

$$\mathbf{F^d} = \exp(\mathbf{F^c} \Delta t), \tag{20}$$

$$\mathbf{G^d} = \begin{bmatrix} \mathbf{G_k^d} \ \mathbf{G_u^d} \end{bmatrix} = (\mathbf{F^c})^{-1}(\mathbf{F^d} - \mathbf{I}^{2N \times 2N})[\mathbf{G_k^c} \ \mathbf{G_u^c}] \tag{21}$$

$$\mathbf{H^d} = \begin{bmatrix} \mathbf{0}^{N \times N} & \mathbf{I}^{N \times N} \end{bmatrix} \tag{22}$$

where N denotes the model's DOF, $\mathbf{0}$ is the null matrix, $\mathbf{I}$ is the identity matrix, and $\mathbf{F^c}$, $\mathbf{G_k^c}$, $\mathbf{G_u^c}$ and $\mathbf{H^c}$ are the matrices of the continuous state space model

$$
\begin{aligned}
\mathbf{F^c} &= \begin{bmatrix} \mathbf{0}^{N \times N} & \mathbf{I}^{N \times N} \\ -\mathbf{J^{-1}K} & -\mathbf{J^{-1}C} \end{bmatrix} \\
\mathbf{G_k^c} &= \begin{bmatrix} \mathbf{0}^{1 \times 2N-1} & 1/J_{Gen} \end{bmatrix}^{\mathrm{T}} \\
\mathbf{G_u^c} &= \begin{bmatrix} \mathbf{0}^{1 \times N} & -i_{DT}/J_{Rot} & \mathbf{0}^{1 \times N-1} \end{bmatrix}^{\mathrm{T}} \\
\mathbf{H^c} &= \mathbf{H^d}
\end{aligned} \tag{23}
$$




For the purpose of simultaneous state and input estimation, the state vector $\mathbf{x}$ is expanded with the unknown input force $\mathbf{u_u}$, yielding the state-space representation with the augmented state vector $\mathbf{x^a} = [\mathbf{x}\ \mathbf{u_u}]^{\mathbf{T}}$.

$$\mathbf{x^a_{n+1}} = \mathbf{Fx^a_n} + \mathbf{G_k u_{k,n}} + \mathbf{w_n}, \tag{24}$$

$$\mathbf{y_n} = \mathbf{Hx^a_n} + \mathbf{v_n}, \tag{25}$$

where the system matrix $\mathbf{F}$, the input matrix $\mathbf{G}$ and the observation matrix $\mathbf{H}$ of the augmented state space model are calculated as follows

$$\mathbf{F} = \begin{bmatrix} \mathbf{F^d} & \mathbf{G^d_u} \\ \mathbf{0^{1 \times N}} & 1 \end{bmatrix} \tag{26}$$

$$\mathbf{G} = \begin{bmatrix} \mathbf{G^d_k} \\ 0 \end{bmatrix} \tag{27}$$

$$\mathbf{H} = \begin{bmatrix} \mathbf{H^d} & \mathbf{0^{N \times 1}} \end{bmatrix} \tag{28}$$

The Kalman filter produces the state estimates $\hat{\mathbf{x}}$ in a two-step algorithm, comprising of the prediction step and the measurement update step.

$$\hat{\mathbf{x}}^{\mathbf{a}}_{\mathbf{n|n-1}} = \mathbf{F}\hat{\mathbf{x}}^{\mathbf{a}}_{\mathbf{n-1|n-1}} + \mathbf{G u_{n-1}}, \tag{29}$$

$$\hat{\mathbf{P}}_{\mathbf{n|n-1}} = \mathbf{F}\hat{\mathbf{x}}^{\mathbf{a}}_{\mathbf{n-1|n-1}}\mathbf{F^T} + \mathbf{Q}. \tag{30}$$

$$\mathbf{M_n} = \hat{\mathbf{P}}_{\mathbf{n|n-1}}\mathbf{H^T}(\mathbf{H}\hat{\mathbf{P}}_{\mathbf{n|n-1}}\mathbf{H^T} + \mathbf{R})^{\mathbf{-1}}, \tag{31}$$

$$\hat{\mathbf{x}}^{\mathbf{a}}_{\mathbf{n|n}} = \hat{\mathbf{x}}^{\mathbf{a}}_{\mathbf{n|n-1}} + \mathbf{M_n}(\mathbf{y_n} - \mathbf{H}\hat{\mathbf{x}}^{\mathbf{a}}_{\mathbf{n|n-1}}), \tag{32}$$

$$\hat{\mathbf{P}}_{\mathbf{n|n}} = (\mathbf{I} - \mathbf{M_n H})\hat{\mathbf{P}}_{\mathbf{n|n-1}}. \tag{33}$$

## 2.8 System identification

System identification methods are applied to continuously update the model properties to ensure the convergence of the virtual model and the physical wind turbine's dynamic behaviour. The rotor inertia, generator inertia, drivetrain torsional stiffness and damping are considered time-variant parameters to reflect long-term changes of the physical wind turbine. The rotor inertia may increase due to the accretion of dirt, moisture and ice, or decrease as a result of leading edge erosion or similar damages. The drivetrain stiffness and damping values are affected by material fatigue and localized faults such as spalls or tooth root cracks. The second line of the equations of motion (Eq. 2.5.2) is used to estimate the parameter set $\theta = [J_{Gen}, c_{DT}, k_{DT}, \alpha_0]$, since the boundary conditions are fully determined here by measurements of the generator torque. The following least-squares optimization problem is then formulated

$$\hat{\theta} = \arg\min_{\theta} ||J_{Gen}\ddot{\phi}_{Gen} - c_{DT}/i_{DT}\dot{\alpha} - k_{DT}/i_{DT}(\alpha - \alpha_0) + T_{Gen}||_2^2 \tag{34}$$



The generator shaft acceleration $\ddot{\phi}_{Gen}$ is obtained by numerical differentiation of the measured SCADA generator shaft speed. The drivetrain torsion defined as $\alpha = \phi_{Rot} - \phi_{Gen}/i_{DT}$ is calculated by numerical integration of the shaft speeds. As a result of the numerical integration of noisy signals, a runaway trend or sensor drift is observed, which is removed via MATLAB's *detrend* function. Furthermore, the initial state $\alpha_0$ of the integrated signal is unknown and therefore added to the parameter set of the optimization problem. The optimization problem is solved for 10 min time sections at each EC using a least-squares solver.

Unfortunately, the same procedure cannot be employed to obtain the remaining parameter, the rotor inertia $J_{Rot}$, since the rotor torque is typically not measured by SCADA systems, which leaves the rotor side equations of motion undefined (Eq. 2.5.2). Operational modal analysis (OMA) techniques are used instead. The first torsional natural frequency $\hat{f}_N$ is estimated using peak finding algorithms in the frequency spectrum of the drivetrain torsion signal $\alpha$. Since the natural frequency is a function of the drivetrain inertia and stiffness, one may solve for the unknown rotor inertia as follows

$$\hat{J}_{eq} = \frac{\hat{k}_{DT}}{(2\pi\hat{f}_N)^2}$$
$$\hat{J}_{Rot} = (1/\hat{J}_{eq} - 1/\hat{J}_{Gen}/i_{DT})^{-1} \tag{35}$$

### 2.9 Fatigue damage

The gear and bearing fatigue damage is based on the gear tooth root stress calculation of ISO 6336 (ISO 6336, 2006) and the nominal bearing life calculation of ISO 281 (ISO 281, 2007). The gear tooth root stress $s$ is determined from the circumferential gear force $F_t$, the flank width $b$, the normal modul $m_n$ and the modification factors $Y$ and $K$ (ISO 6336, 2006)

$$s = \frac{F_t}{bm_n} Y_S Y_F Y_\beta Y_B Y_{DT} K_A K_V K_{F\beta} K_{F_\alpha} K_\gamma \tag{36}$$

The pendant for bearings is the equivalent dynamic load $P$ that is defined for for cylindrical roller bearings (CRB) and tapered roller bearings (TRB) as follows (ISO 281, 2007):

$$\text{for CRB:} \quad P = F_{rad} \tag{37}$$

$$\text{for TRB:} \quad P = \begin{cases} F_{rad} + Y_1 F_{ax}, & \text{if } F_{ax}/F_{rad} \le e \\ 0.67 F_{rad} + Y_2 F_{ax}, & \text{otherwise} \end{cases} \tag{38}$$

where $Y_1, Y_2, e$ are bearing-specific parameters.

The load duration distribution (LDD) method is used as stress cycle counting method for components in rotating machinery that experience cyclic loading due to entering and exiting the load zone (Nejad et al., 2014). The LDD method counts one stress cycle per shaft revolution and distributed the cycles $n_i$ into 64 bins of increasing stress range. The permissible stress cycles $N_i$ for each stress range is modelled with S-N curves for gear tooth root fatigue

$$N_i = K_c\, s_i^{-m} \tag{39}$$




where $m = 6.225$ and $K_c = 10^{24.744}$ (Nejad et al., 2014), and the nominal bearing life equation for bearing fatigue (ISO 281, 2007)

$$N_i = 10^6 \left( \frac{C}{P_i} \right)^m \tag{40}$$

where $C$ is the basic dynamic load rating and $m = 10/3$ for roller bearings.

The short-term fatigue damage is then calculated for 10 min time sections by summation of all stress range bins

$$D^{ST} = \sum_i n_i / N_i \tag{41}$$

The long-term fatigue damage $D^{LT}$ for the nominal life time of 20 years is extrapolated from the short-term fatigue damage by weighting with the wind speed probability density function $f(u_k)$. A representative wind speed distribution measured at

Anholt, Denmark is selected.

$$D^{LT} = \frac{20 \text{ year}}{10 \text{ min}} \sum_k f(u_k) D_k^{ST} \tag{42}$$

## 3   Results and discussions

### 3.1   Choice of uncertainty distribution

The first step in the statistical analysis of the uncertainty in DTs is the identification of the distribution types, which are of

importance in reliability and risk assessment studies. A common assumption is to use log-normal distributions for representing model uncertainties (Nejad et al., 2014; Dong et al., 2020). The numerical results of the measurement, state estimation, system identification and model uncertainty are fitted with fourteen different statistical distributions and ranked according their goodness of fit given by the coefficient of determination $R^2$. Fig. 4 shows the $R^2$-values of the six best performing distributions aggregated for all EC of the 5 MW case study. The results are inconclusive as to which distribution is best suited, but it can be

stated that the log-normal distribution yields a reasonable fit of $R^2 > 0.9$ for all types of uncertainty in DTs. The further statistical analysis is continued with log-normal distributions to maintain the comparability with other publications. Log-normal distributions are defined as $X = \exp(\mu + \sigma Z)$, where $Z$ is a standard normal variable. Note that the parameters $\mu$ and $\sigma$ differ from the distribution's mean and standard deviation. The findings of this study are summarized in table 4 and discussed in the following sections

### 3.2   Measurement uncertainty

The first source of uncertainty in the proposed load and fatigue monitoring approach originates from the the low temporal resolution of the SCADA data input. Typical SCADA systems operate with sampling frequencies of 1 Hz, but store the data only as 10 min averages, which has already been identified as a limiting factor for monitoring approaches. The generator torque reportedly has the fastest decaying autocorrelation out of all SCADA signals, which results in a large loss of information when





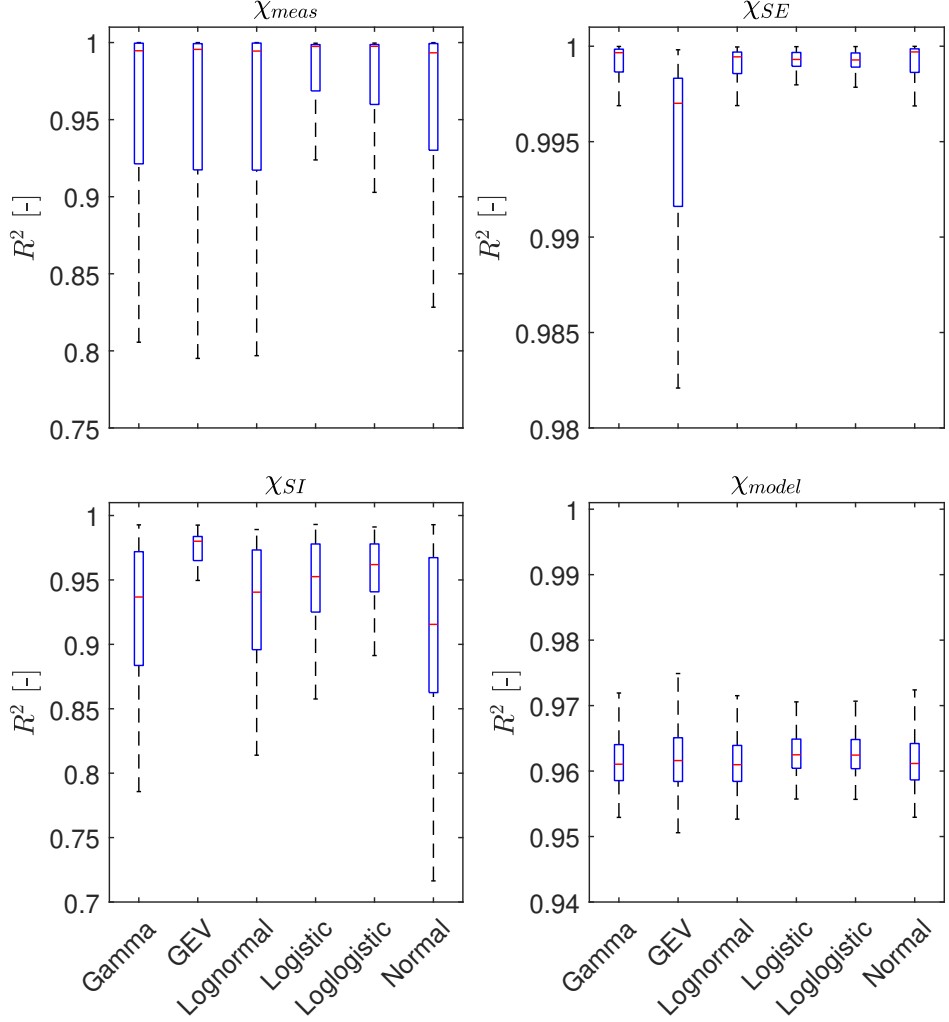

**Figure 4.** Goodness of fit of different distribution shapes for the measurement, state estimation, system identification and model uncertainty, aggregated for all ECs of the 5 MW case study.

using time averaged signals (Gonzalez et al., 2019). This motivated efforts in the industry to adopt high frequency (1 Hz) SCADA systems; however, even a sampling frequency 1 Hz is arguably insufficient to fully capture drivetrain dynamics, since the first torsional natural frequency and internal excitation frequencies such as gear meshing frequencies lie well above the Nyquist frequency of 0.5 Hz. The effects of this are illustrated in Fig. 5, which shows the $\sigma_{\chi meas}$-parameter of the fitted measurement uncertainty $\chi_{meas}$ resulting from either 1 s or 10 min averaging of the generator torque input. The measurement

uncertainty of 10 min data is particularly high below rated wind speed and reaches values of up to 0.75 near cut-in wind speed. In wind turbines with variable-speed controllers this operational regime is characterized by a high variance in the drivetrain torque, which is not reflected in 10 min averaged data. The uncertainty of 1 Hz data only amounts to only about 0.03 for most

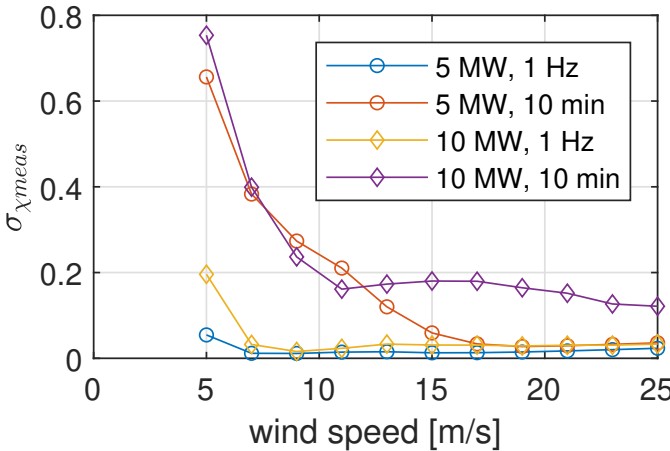

**Figure 5.** $\sigma$-parameter of fitted log-normal distributions for measurement uncertainty as a function of wind speed.

operational conditions with the exception of cut-in wind speeds, which suggests that this resolution is sufficient to observe low-frequency ($< 0.5\ Hz$) load variations due to the wind speed volatility. The remaining uncertainty is related to neglecting higher frequency dynamics such as torsional drivetrain modes. Based on these results it a measurement resolution of at least 1 Hz is recommended for load and fatigue damage monitoring in wind turbine drivetrains.

### 3.3 State estimation uncertainty

The second source of uncertainty is also related to the limitations of the SCADA measurements, in that the rotor torque is typically not measured and must be estimated indirectly using the augmented Kalman filter as joint input-state estimation method. The error in the estimated rotor torque is described by the state estimation uncertainty $\chi_{SE}$. Fig. 6 shows the $\sigma$-parameter of $\chi_{SE}$ from numerical case studies with the 5 MW and 10 MW model at different ECs. Particularly high uncertainty is observed around cut-in wind speeds, which can be attributed to start-up and shut-down effects. At normal power generation the uncertainty is limited to values of 0.07 and 0.04 for the 5 MW and 10 MW turbine, respectively. A slightly higher error is observed with the 5 MW model, which is also apparent in the frequency spectra and time series shown in Fig. 7. The rotor torque estimates for the 10 MW turbine show a good agreement in the low-frequency range and at the peaks of the first torsional natural frequency (2.08 Hz). For the 5 MW turbine, on the other hand, the rotor torque is underestimated at the first torsional natural frequency (1.7 Hz) and at higher order modes.

### 3.4 System identification uncertainty

The third source of uncertainty originates from the aleatory uncertainty of the system properties, for instance, the rotor inertia may vary due to ice accretion or leading edge erosion and the drivetrain stiffness may decrease due to tooth root cracks or spalling. System identification methods are applied to detect these changes and update the model parameters accordingly, nonetheless, a small uncertainty of epistemic nature remains due to method limitations, referred to as the system identification



**Figure 6.** $\sigma$-parameter of fitted log-normal distributions for state estimation uncertainty as a function of wind speed.

uncertainty $\chi_{SI}$. The system identification uncertainty in the parameters $J_{Rot}, J_{Gen}, k_{DT}, c_{DT}$ is investigated in numerical case studies with the 5 MW and 10 MW models. The numerical results are shown in Fig. 8 as the $\mu-$ and $\sigma$-parameter of the
fitted log-normal distributions across all ECs. The uncertainty in the inertia and stiffness parameter estimation shows similar behaviour. Local maxima in the bias and variance are observed near cut-in (5 m/s) and near rated wind speeds (11-13 m/s), while the minimum is located at cut-out wind speed (25 m/s). It appears that the quasi-stationary conditions in the torque controlled operational regime above rated wind speeds are conducive to accurate parameter estimation, while the transient dynamics at rated wind speeds due to activation and deactivation of the pitch controller introduce higher estimation errors.
Contrary to the inertia and stiffness estimates, the damping parameter estimates show significantly higher uncertainty reaching values of up to $\sigma > 0.55$. This finding is in agreement with recent studies on drivetrain model validation, where it is reported that the estimation of damping values by OMA techniques is challenging due to the low parameter sensitivity (Vanhollebeke et al., 2015). The damping parameter has outside of the resonance area, at the considered operational conditions a small influence on the dynamic response.


### 3.5 Model uncertainty

Lastly, the model uncertainty $\chi_{model}$ is investigated, which characterizes the uncertainty in the calculated bearing and gear loads due to modelling errors and the complexity reduction of the ROMs. The discussion is divided into a frequency analysis (Sec. 3.5.1), the analysis of the model bias (Sec. 3.5.2) and the analysis of the dynamic model error (Sec. 3.5.3).

#### 3.5.1 Characterization of drivetrain dynamics

A frequency analysis of the simulated drivetrain loads is conducted to identify which aspects of the drivetrain dynamics the ROMs are able to represent well and which aspects are sources of uncertainty. The drivetrain dynamics can be generally

**Figure 7.** True and estimated rotor torque $T_{rot}, \hat{T}_{rot}$ using joint state-input estimation methods. Shown are the PSD frequency spectrum and the time series at EC8.

characterized as dynamic responses to a variety of both internal and external excitations. These excitations can be further differentiated into torque and non-torque loads, i.e lateral forces and bending moments (Tab. 3.5.1).

External excitations are mainly the result of aerodynamics and are prevalent at low frequencies. Aerodynamic imbalance is present in healthy conditions due to turbulence, wind shear, the vertical wind profile and the rotor axis tilt, or caused by faulty yaw and pitch misalignment. This results in periodic load variations in the rotor torque, thrust and bending moments at the rotor frequency 1P (Mehlan et al., 2023). The tower shadow is also known to induce similar torque and non-torque excitations at the blade passing frequency 3P.

The system boundaries of the drivetrain models cut through the rotor hub and the yaw bearing, hence, all structural dynamics of the blades and the tower are considered as external excitations. These are simulated with the global aeroelastic models and the resulting main shaft loads and tower motions are applied as boundary conditions in the drivetrain models. The deformation



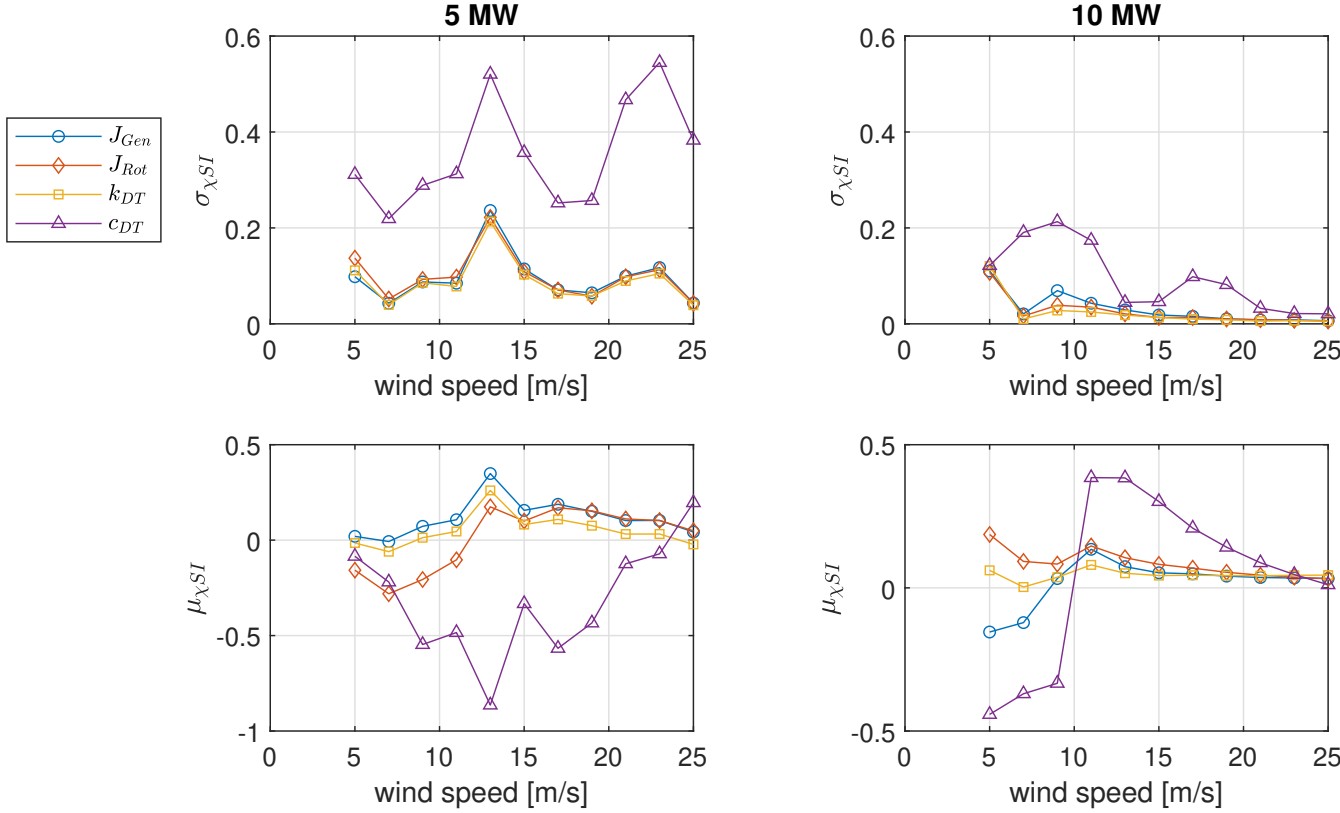

**Figure 8.** $\mu$- and $\sigma$-parameter of fitted log-normal distributions for system identification uncertainty as a function of wind speed.

of the blades with edgewise bending modes translates to torque excitations at the main shaft, while flapwise bending modes cause primarily non-torque excitations. Similarly, fore-aft and side-side tower bending introduces excitations in the thrust and 360 bending moments.

Internal excitations are caused by periodic changes of component stiffnesses and occur generally at much higher frequencies. Gear mesh excitations are a result of the changing number of tooth contacts during one meshing cycle. Gear meshing primarily results in periodic variation of the transmitted torque, but may also have non-torque components in helical gear stages. Bearing excitations are caused by roller elements passing the load zone and result in non-torque excitations at the ball passing frequen-365 cies. Further internal excitations are observed at the planet carrier rotational frequencies. Shaft misalignment, mass imbalance or non-torque loading may result in bending of the flexible planet carrier and in skewing of the load distribution between planets, such that each planet bearing experiences periodic load changes during one planet carrier revolution.

The characteristic excitations are observable in the power spectral densities (PSD) of the bearing loads (Fig 9). Shown are the simulated bearing loads at each gear stage for EC8 (17 m/s) using the FOM and the rigid and flexible ROM. The rigid ROM 370 exhibits a good agreement in the lowest frequency range ($< 1\,Hz$) governed by wind and wave load excitations, but generally underestimates higher frequency dynamics, as it is only considering rigid body modes. The flexible ROM achieves more




**Table 2.** Type of excitations and characteristic frequencies in wind turbine drivetrains

|  |  | Torque | Non-Torque |
|---|---|---|---|
| **External** | | Aerodynamic imbalance ($f_{1P}$) | Aerodynamic imbalance ($f_{1P}$) |
| | | Tower shadow ($f_{3P}$) | Tower shadow ($f_{3P}$) |
| | | Blade edgewise modes ($f_N$) | - |
| | | - | Blade flapwise modes ($f_N$) |
| | | - | Tower bending modes ($f_N$) |
| **Internal** | | Planet carriers ($f_{plc}$) | Planet carriers ($f_{plc}$) |
| | | Gear meshing ($f_{gm}$) | Gear meshing ($f_{gm}$) |
| | | - | Bearings ($f_{bpf}$) |

accurate load estimates by inclusion of the first torsional drivetrain mode. It is able to match the peaks of external excitations such as the first collective edgewise blade bending mode ($f_{N1}$) and higher order modes. The internal dynamics are captured reasonably well with a good agreement in the second stage gear meshing frequency ($f_{gm2}$). However, some discrepancies

remain in the first stage gear meshing frequency peak ($f_{gm1}$) and in the planet carrier excitations ($f_{plc1}$, $f_{plc2}$) visible at the first and second stage planet bearings (PL-A, IMS-PL-A). These suggest the presence of non-torque loads at the planet carriers. The investigated 5 and 10 MW drivetrain models are designed with a four-point main bearing suspension, where it is generally assumed that all non-torque loads of the rotor are fully compensated by the main bearings, but it appears that this is not the case and that non-torque loads partially propagate further downwind into the drivetrain. The results showcase the limitations

of torsional ROMS and suggest that a significant source of uncertainty originates from neglecting planetary carrier bending modes.

### 3.5.2   Model bias

The focus of the statistical analysis lies first on the model bias, which is quantified by the $\mu$-parameter of the fitted model uncertainty distribution. Values of greater than one represent consistent underestimation of drivetrain loads by the ROMs and

vice versa for values smaller than one. Shown in Fig. 10 are the model biases of the rigid and flexible ROM in numerical and experimental case studies. The field measurements are only available for the HSS-A bearing. The highest biases are observed near cut-in wind speeds (5 m/s), which can be associated with start-up and shut-down effects. At higher wind speeds ($> 7 m/s$) the environmental conditions have a marginal influence on the model bias. Significant biases of up to 0.46 are observed at the high-speed gear stage. The loads at the upwind HSS-A and IMS-A bearings are consistently underestimated, while the

loads at the downwind HSS-B and IMS-B bearings are overestimated. One reason for these discrepancies could lie in the physical simplifications of the ROMs, which reduces the gear contact force to a singular vector along the line of action. The load distribution along the gear flank is not considered and thus the bending moments resulting from inhomogeneous load distributions are neglected. Other authors introduce a "twist stiffness" perpendicular to the circumferential gear meshing stiffness to account for the load distribution (Eritenel and Parker, 2012). However, in this approach the solution requires

knowledge of gear and bearing stiffness parameters, which are difficult to determine and validate in practice. Another factor



eawe
european academy of wind energy

WIND
ENERGY
SCIENCE
DISCUSSIONS

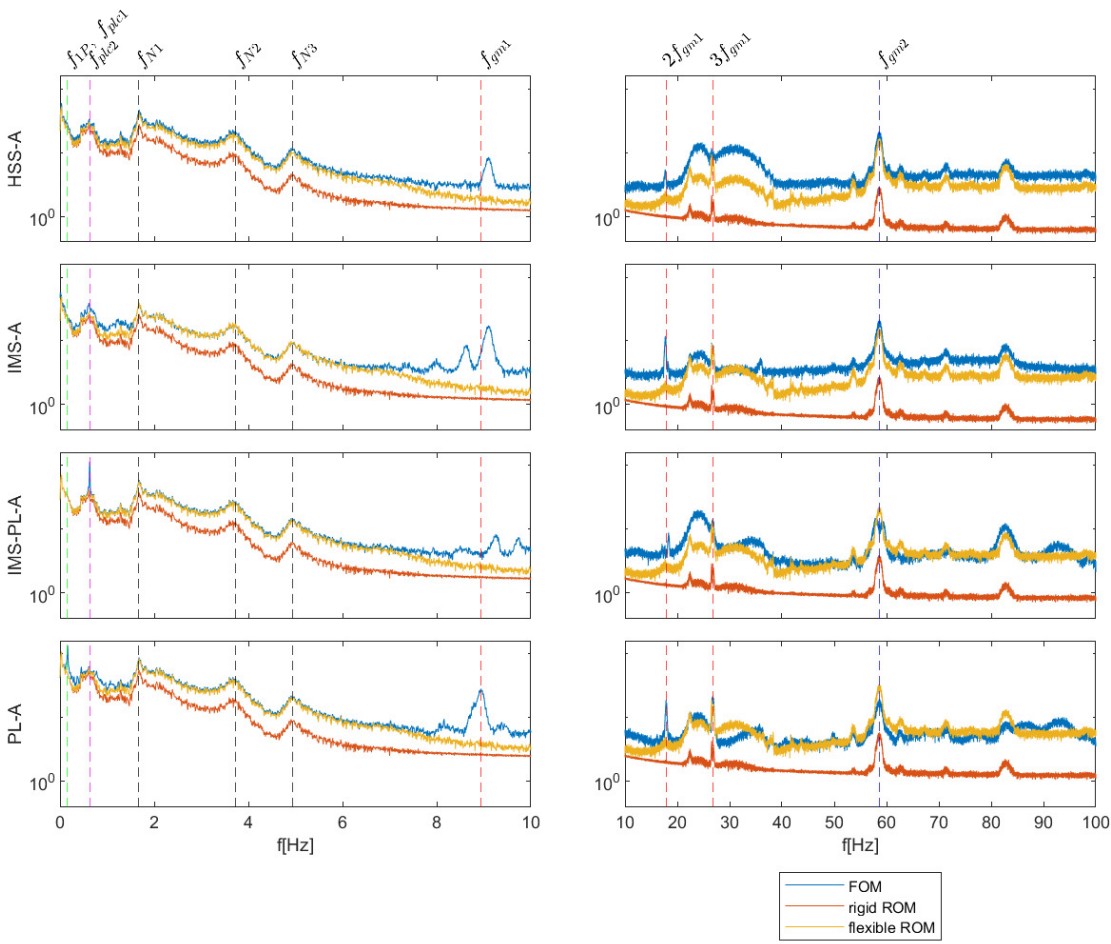

**Figure 9.** Power spectral densities of bearing radial loads simulated with the 5 MW FOM, rigid ROM and flexible ROM at EC8.

could be the assumption of open-ended shafts that do not allow the transfer of non-torque loads. In the FOMs this is not the case, since the generator coupling at the HSS and the sun-planet gear contact at the IMS allow the transfer of shear forces. These could skew the HSS and IMS bearing loads and further contribute to the model bias. The persistence of model biases in such analytical ROMs is further supported with field measurements of the DOE 1.5 MW turbine. The measured model bias is
independent of the EC and amounts to about 0.15, which is of similar magnitude as the values of the numerical case studies.

### 3.5.3  Dynamic error

The $\sigma$-parameter of the of the fitted uncertainty distributions indicates how well the ROMs capture drivetrain dynamics compared the FOM. As depicted in Fig. 11, the $\sigma$-parameter is positive for all considered cases, which suggests that the ROMs



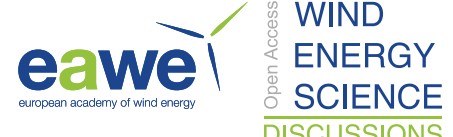



**Figure 10.** $\mu$-parameter values of fitted log-normal distributions for model uncertainty as a function of wind speed.

generally underestimate the load dynamics. The uncertainty distributions show similar trends across all bearing and gear types. The highest values are observed near cut-in wind speeds (5 m/s), followed by a steep decline to the global minimum at 9 m/s and a gradual progressive trend towards cut-out wind speeds (25 m/s). Similarly to the high model bias, the high uncertainty at cut-in wind speeds can be attributed to start-up and shut-down effects. The progressive trend can be attributed to aerodynamic non-torque loads transferred from the rotor into the drivetrain. While the torque is controlled to rated conditions above rated wind speed, the non-torque loads, in particular pitch and yaw bending moments, continue to increase with higher wind speeds (Mehlan et al., 2022b). These can excite non-torsional modes of the drivetrain, in particular planet carrier bending modes (see Sec. 3.5.1), which the purely torsional ROMs do not account for.







**Figure 11.** $\sigma$-parameter of fitted log-normal distributions for model uncertainty as a function of wind speed.

The flexible ROM appears to capture the drivetrain dynamics to a much higher degree than the rigid ROM resulting in lower uncertainty values across all bearing and gear locations. The largest differences are observed above rated wind speed, where the excitation of the first drivetrain torsional mode becomes increasingly more energetic. Below rated wind speed the relative improvement is much lower, since in this operational regime the drivetrain dynamics are governed by rigid-body modes. The uncertainty based on field measurements shows a similar trend and order of magnitude and supports the previous findings of the numerical case studies.



## 3.6 Long-term fatigue damage error

The use case of long-term fatigue damage monitoring is considered to assess the impact of the uncertainties in the DT framework. Three scenarios are hereby considered with increasing resolution of SCADA measurements, ranging from 10 min, 1 Hz to 200 Hz. The resolution of 10 min and 1 Hz limits the DT model to the rigid torsional ROM, since the first torsional natural frequency lies above the Nyquist-frequency, while the case of 200 Hz measurements allows the application of the flexible ROM. The long-term fatigue damage is calculated by weighting the short-term fatigue damage of each EC with the wind speed

distribution.

As shown in Fig. 12, the contribution of wind speeds near cut-in (3-7 m/s) to long-term fatigue does not exceed 2% due to the low probability of such wind speeds in addition to small aerodynamic loads. The small contribution suggests that the high uncertainty observed at cut-in wind speeds due to start-up and shut-down effects (Sec. 3.5.2) has a negligible impact. The highest contribution have wind speeds of 13 m/s, where model and measurement uncertainty are fortunately near their minima.

The relative error in long-term fatigue damage for each of the scenarios is shown in Fig. 13. The long-term fatigue damage is generally underestimated by the DTs due to underestimation of the load amplitudes. It should be noted that the error in the bearing and gear load estimates is amplified by exponentiation with the S-N curve exponent of 10/3 and 6.225, respectively. Hence, the gear fatigue damage error tends to be larger due to the larger exponent.

The first scenario with 10 min SCADA data results in relative errors of up to -44.4% in the gear fatigue damage and up to

-15.9% in the bearing fatigue damage due to the high measurement uncertainty $\chi_{meas}$ (Sec. 3.2). The resolution is insufficient to capture neither the low-frequency aerodynamics nor the high-frequency internal drivetrain dynamics. The second scenario with 1 Hz data yields significantly smaller relative errors limited to -11.2% and -6.6% in the gear and bearing fatigue damage, respectively. In this case, the rigid ROM is able to represent low frequency load variations due to wind and wave excitations, but is limited with respect to higher frequency internal dynamics dynamics. The third scenario with 200 Hz measurements and

the two DOF flexible ROM results in only marginally lower fatigue damage errors of -9.7% and -5.5%, which showcase the trade-off of increasing the model fidelity. While the addition of a torsional DOF in the flexible ROM significantly reduces the modelling errors and the model uncertainty $\chi_{model}$ (Sec. 3.5.3), it introduces one unknown variable in the rotor torque and four unknown parameters in the rotor intertia, generator inertia, drivetrain stiffness and damping. The estimation of the rotor torque and the parameters by inverse methods cause additional uncertainty $\chi_{SE}$, $\chi_{SI}$ (Sec. 3.3 and 3.4), which partially diminish the

benefit of the lower model uncertainty.

## 4 Conclusions

This paper presents a systematic assessment of the uncertainty of DTs for load and fatigue damage monitoring in wind turbine drivetrains. Numerical studies with the NREL 5 MW and DTU 10 MW reference turbines and experimental studies with the DOE 1.5 MW research turbine were conducted to assess the uncertainty of different DT elements and their impact of long-

term fatigue damage. The measurement uncertainty in the SCADA data input $\chi_{meas}$, the uncertainty in the state estimation and system identification methods $\chi_{SE}$, $\chi_{SI}$, and the model uncertainty of the drivetrain ROMs $\chi_{model}$ were investigated and

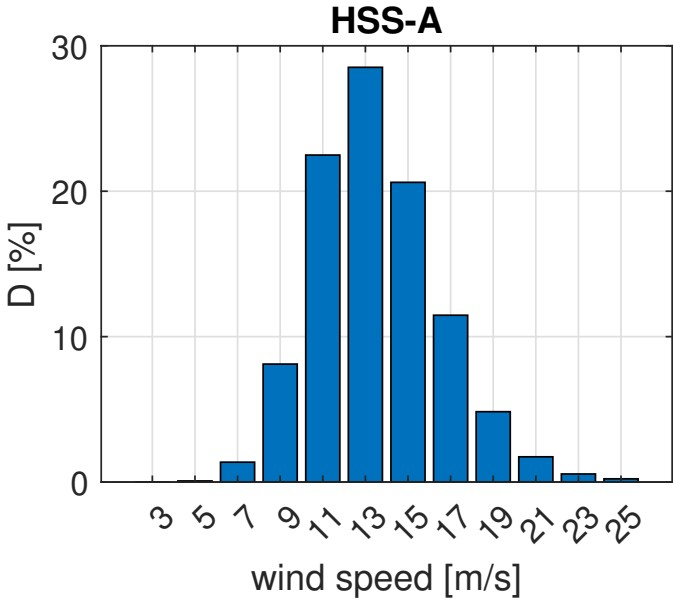

**Figure 12.** Contribution of each wind speed bin to long-term fatigue damage for the example of the 5 MW HSS-A bearing.

quantified using log-normal distributions (Tab. 4)

The investigation of the measurement uncertainty revealed a significant loss of information by using 10 min averaged SCADA data. The measurement resolution is insufficient to observe the low frequency drivetrain load dynamics due to the wind speed and rotor torque volatility, which resulted in maximum uncertainty of $\sigma = 0.75$ and long-term fatigue damage errors of up to -44.4% in the gears and -15.9% in the bearings. The results strongly suggest the use of high-frequency SCADA data with a resolution of at least 1 Hz for fatigue monitoring purposes.

The second source of uncertainty is identified in the state estimation method, the augmented Kalman filter, that is applied to match the dynamic state of the DT model with the physical wind turbine based on real-time data streams. The challenge lies in estimating the rotor torque, which is not measured directly and must be estimated by the Kalman filter. The Kalman filter tends to underestimate the rotor torque at the first torsional natural frequency, which results in a uncertainty ranging from $\sigma = 0.03...0.07$ at normal operational conditions ($> 5m/s$).

The third source of uncertainty originates from the aleatory uncertainty of the system properties. Inertia, stiffness and damping values may vary over the turbine's life cycle as a result of faults, material degradation or part replacement. System identification methods are applied to detect these changes and update the model parameter accordingly. The uncertainty in the parameter estimates is particularly high at cut-in and near rated wind speeds ($\sigma < 0.22$) due to transient dynamics and the high variance in the drivetrain torque, while the lowest uncertainty is observed in the torque controlled regime above rated wind speed ($\sigma > 0.006$). Furthermore, it is observed that the estimation of the drivetrain torsional damping is significantly more inaccurate than inertia and stiffness parameters ($\sigma < 0.22$). This is likely due to the low sensitivity of the damping parameter with respect to the drivetrain torsional dynamics at normal power production.


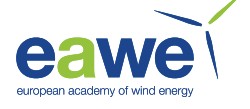
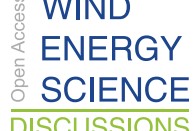

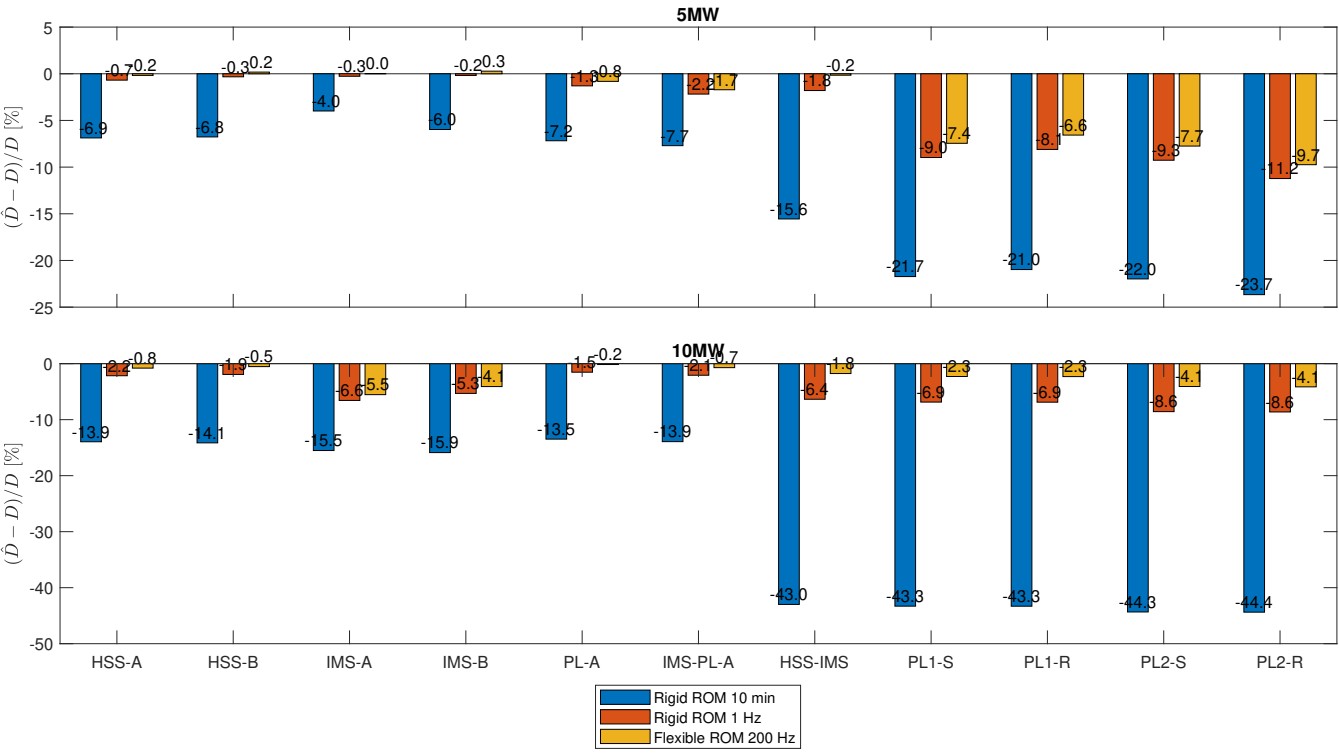

**Figure 13.** Relative error [%] in long-term bearing and gear fatigue damage

Lastly, the epistemic model uncertainty due to the ROMs' limitations is investigated. ROMs with one or two torsional DOFs are used as DT models due to their lower computational loads in real-time monitoring applications, their lower validation costs, and the limited observability of non-torsional dynamic states with the available SCADA measurements. One DOF rigid ROMs are only able to match the dynamics in the lowest frequency range ($< 1\ Hz$) governed by wind and wave load excitations, while two DOF flexible ROMs better capture the dynamic drivetrain response to higher frequency internal excitations such as gear meshing. Remaining limitations are observed in capturing non-torsional dynamics, in particular the bending dynamics of the first and second stage planet carriers. The uncertainty in the load estimates of the flexible ROM is noticeably smaller, however only a small improvement with respect to the fatigue damage estimates is observed ($-6.6\%$ to $-5.5\%$). While the addition of a torsional DOF in the flexible ROM significantly reduces the modelling errors, it introduces additional unknown variables and parameters with associated uncertainties that partially diminish the benefit of lower modelling uncertainty.

The presented study contributes to a deeper understanding of the uncertainty in DTs for load and fatigue monitoring. The reported uncertainty distributions may be used in reliability studies, in risk assessment and the derivation of safety factors, or assist in the decision processes on the model fidelity and the sensor measurement resolution.





**Table 3.** Summary of the uncertainty quantification in DTs.

| Uncertainty | distribution | $\mu$-parameter | $\sigma$-parameter |
|---|---|---|---|
| Measurement $\chi_{meas}$ | log-normal | 0 | 0.02...0.75 |
| State estimation $\chi_{SE}$ | log-normal | 0 | 0.03...0.07 |
| System identification $\chi_{SI}$ | log-normal | $-0.86..0.38$ | 0.01...0.55 |
| Model $\chi_{model}$ | log-normal | $-0.45...0.28$ | 0.01...0.17 |

*Data availability.* The data will be made available upon request.

*Author contributions.* **Felix Mehlan**: Methodology, Software, Investigation, Writing - Original Draft, **Amir R. Nejad**: Conceptualization, Writing - Review & Editing, Funding acquisition

*Competing interests.* The co-author Amir R. Nejad is a member of the editorial board of Wind Energy Science Journal.

*Acknowledgements.* The authors wish to acknowledge financial support from the Research Council of Norway through InteDiag-WTCP
project (Project number 309205).





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
