# Peer review of "On the Modelling Errors of Digital Twins for Load Monitoring and Fatigue Assessment in Wind Turbine Drivetrains"

_Wind Energy Science, 2024_

## Author Response (AR1)

**Rebuttal**

The reviewers' constructive comments and contributions to improve our work is greatly appreciated. We carefully read the comments and would like to present our response here. We would like to address the following concerns that are related to the definition of the uncertainty.

Reviewer 1:

*I have some doubts regarding the conceptual framework. The authors quantify the uncertainty of the employed models by comparing numerical simulations (with high fidelity models and 200 Hz of frequency) against other numerical simulations (which have lower frequency, or employ Kalman filters, or simulate a dynamic behavior through reduced models). For me, it is slightly misleading to call this "uncertainty". The uncertainty is something related to a process of measurement. Sincerely, I would rather call it information loss, or something like similar.*

*Furthermore, I am not convinced by the way the authors define the various uncertainties. For example, above Equation 3, the authors say that the system identification uncertainty is defined as the ratio of the true system parameter to the estimated parameter set. I do not agree. An uncertainty is a difference with respect to a true parameter. One might consider the relative uncertainty, which is the ratio of the difference with respect to a true parameter to the true parameter itself. None of these have the form of true / estimated value. I suggest elaborating on this point and presenting the problem in a more consistent way.*

Reviewer 2:

*However, I do agree with the first reviewer's opinion that the "uncertainty" should be defined as the difference between simulated results and real measured data. Hence I suggest revising the conceptual framework as a study on the information loss under different simulation conditions.*

Our response:

Our initial definition of the uncertainty was adopted from the field of structural reliability-based design, where the model uncertainty for example in the fatigue damage calculation $\chi$ is the product of different model uncertainties in the computational chain such as the aerodynamic model $\chi_{aero}$ and the drivetrain model $\chi_{dyn}$ (see more details here https://doi.org/10.1016/j.ijfatigue.2013.11.023)

$$\chi = \chi_{aero} \cdot \chi_{dyn} \cdot \cdots$$

The overall uncertainty is represented in the fatigue failure function $g(X)$ as follows

$$g(X) = \Delta - \chi^m D_{LT} \geq 0$$

Where $\Delta = 1$ is the failure limit and $D_{LT}$ is the long-term fatigue damage. In this framework it is sensible to define the uncertainty as the ratio of the estimated to the true loads.

$$\chi := \frac{F_{true}}{F_{est}}$$

However, we agree that this formulation can be confusing outside of the area of reliability-based design and therefore we adopt the relative error $e$ as our uncertainty metric in the revised paper.

$$e := \frac{F_{est} - F_{true}}{F_{true}}$$

Furthermore, normal distributions rather than log-normal distributions are selected, as they better characterize the relative error and are more comprehensive.

In addition, we discussed internally whether our analysis characterizes the model uncertainty or only the modelling errors/information losses in the fatigue damage calculation, since the reference values are not obtained from measurements, but rather from simulations and therefore do not represent the "true" values. In our opinion, the simulation results from high-fidelity MBS models capture more realistic drivetrain behavior to a high degree and are therefore suitable to be used as reference values. The analysis using simulation results as the ground truth can thus provide a good estimate of the actual uncertainty that is to be expected in the field. However, to avoid misunderstanding we decided to reformulate our approach and use the terms "modelling and estimation errors" instead of "uncertainty" and emphasize that the errors are in relation to high-fidelity simulation models. The manuscript was edited accordingly, and a summary of the changes is listed below.

Summary of changes to the manuscript

1) Emphasized that the "true" values are simulation results from high-fidelity models

2) Reformulated and redefined the "uncertainty" $\chi = \frac{true}{estimated}$ as "relative modelling/estimation error" $e = \frac{estimated - true}{true}$.

3) Changed the distribution shape from log-normal to normal.

4) Updated all figures to display relative errors in %.

5) Fixed one issue related to very high errors at cut-in wind speeds from shut-down and start up effects by filtering for normal power production.

---

## Author Response (AR2)

**Rebuttal**

We would like to extend our gratitude to the reviewers for their thorough evaluation of our manuscript and for providing insightful feedback. Your constructive comments and suggestions have been invaluable in guiding the revisions and improving the quality of our work. We have carefully considered each point raised and have made the necessary revisions to address the concerns. Below, we provide a detailed response to the reviewers' comments.

**Reviewer #1**

No comments were provided by reviewer #1

**Reviewer #2**

1. Minor comments with respect to grammar issues of the paper:

1) In the paragraph below line 35, two 'the' were used in "...however, there remain research questions on the sources and the magnitude of the the virtual

measurements' uncertainty."

2) In e.q. (22), one parentheses is in the wrong place, please check.

3) In the sentence above line 320, it says "Based on these results it a measurement resolution of at least 1 Hz is recommended for load and fatigue damage monitoring in wind turbine drivetrains", here the 'it' shall be removed? please check.

4) In line 456, Tab. 4 is not given in the paper, which shall be Tab.3? Please check and update.

*The grammar mistakes have been corrected.*

2. In line 196, it says "...Nonetheless, the full wind spectrum is covered..". How to understand this? A brief explanation shall be given here for a better understanding.

*The amount of data for each wind speed bin is sufficient to characterize the dynamic behaviour in the full range of operational conditions, which allows the comparison with simulated data from the numerical case studies.*

3. For dynamic analysis, how to determine the damping values is usually a challenging task. For the numerical simulations performed in the paper (global analysis and local drive train analysis), what kind of damping models were applied? and what are the damping values commonly used? It is recommended to include more detailed information about the estimation of damping in the paper.

*A description of the damping models was added to Sec. 2.4. The full-order models comprise a large number of components with different damping models. Bearing and gears are typically modelled as spring-damper connections with stiffness proportional damping of about 1% and 0.1%, respectively. Flexible bodies such as the main shaft are modelled as condensed FE models, which are derived from FE models by modal reduction techniques. Here, modal damping of 2% is considered. The reduced-order model considers only one torsional damping constant, as described in Sec. 2.5. To allow the comparison of ROM and FOM, the equivalent torsional damping constant of the FOMs is adopted from Wang et al. (2020) and Nejad et al. (2016) and verified with decay tests. This value is considered the ground truth and compared with the damping parameter estimates obtained by system identification methods.*